# Growth Performance, Rumen Fermentation and Inflammatory Response on Holstein Growing Cattle Treated with Low and High Non-Fibrous Carbohydrate to Neutral Detergent Fiber Ratio Pelleted Total Mixed Ration

**DOI:** 10.3390/ani12081036

**Published:** 2022-04-15

**Authors:** Yinyin Chen, Xiaoxiao Gong, Yinghao Huang, Maocheng Jiang, Kang Zhan, Miao Lin, Guoqi Zhao

**Affiliations:** College of Animal Science and Technology, Yangzhou University, Yangzhou 225009, China; chenyinyin90s@163.com (Y.C.); 18762329160@163.com (X.G.); huangyinghao798@163.com (Y.H.); jmcheng1993@163.com (M.J.); kzhan@yzu.edu.cn (K.Z.); linmiao@yzu.edu.cn (M.L.)

**Keywords:** low and high NFC/NDF, pelleted total mixed ration, growth performance, rumen fermentation, inflammatory response, Holstein growing cattle

## Abstract

**Simple Summary:**

Total mixed rations (TMR) have become the usual feeding method for cattle, since being brought into ruminant feeding systems. Though there are numerous benefits of TMR, it also possesses some drawbacks, especially feed sorting. This has resulted in the recent tendency to provide fully pelleted TMR to sheep and goats in some regions of the world. However, the application of pelleted TMR treated with different non-fibrous carbohydrate (NFC) to neutral detergent fiber (NDF) ratio on Holstein growing cattle was unknown. This research was carried out to investigate the influences of low and high NFC to NDF ratio pelleted TMR on growth performance, rumen fermentation and inflammatory response in Holstein growing cattle. The results demonstrated that low NFC/NDF (1.07) pelleted TMR can improve the growth performance, reduce feed costs and elevate the digestibility of DM, NDF as well as ADF of Holstein growing cattle. However, high NFC/NDF (1.71) pelleted TMR has adverse effects on rumen fermentation and inflammatory response compared with low NFC/NDF (1.07). Under the condition of this experiment, low NFC/NDF (1.07) pelleted TMR is recommended when applied for Holstein growing cattle fattening.

**Abstract:**

Twenty-four healthy castrated male Holstein growing cattle, with similar body weight (301 ± 11.5 kg), were enrolled in this study and randomly allocated into two groups (12/pen). Holstein growing cattle in the LPT (low NFC/NDF pelleted TMR) group were fed basal pelleted TMR with a low NFC/NDF ratio (NFC/NDF = 1.07), while the HPT (high NFC/NDF pelleted TMR) group were fed with a high NFC/NDF ratio diet (NFC/NDF = 1.71). The results showed that: (1) Body measurements were found to be increased for the LPT group (*p* < 0.05); compared with the HPT group, feed intake to gain ratio and feed cost in the LPT group were decreased by 12.24% and 15.35%, respectively (*p* < 0.01). Compared with the HPT group, the LPT group tended to increase chest girth. (2) Digestibility of DM and NDF in the LPT group was higher (*p* < 0.05) than in the HPT group, being increased by 3.41% and 4.26%, respectively, and increased digestibility of ADF in the LPT group was significant (*p* < 0.01). (3) The daily feed consumption of NDF in the LPT group was higher than that in the HPT group and the daily rumination time and chewing time in the LPT group were longer than that in the HPT group (*p* < 0.05). (4) Compared with the LPT group, the parameter of pH, microbial protein and acetate: propionate (*p* < 0.05) in the HPT group were decreased by 8.57%, 12.46% and 23.71%, respectively. In contrast, the concentration of total volatile fatty acids, acetate and propionate were higher (*p* < 0.05) in the HPT group, and increased by 13.49%, 19.59% and 52.70%, respectively. (5) Compared with the LPT group, rumen fluid in the HPT group diet up-regulated the mRNA expression levels of BRECs pro-inflammatory factor IL-1β and TNF-α (*p* < 0.05), and meanwhile, up-regulated the mRNA expression levels of BRECs pro-inflammatory factor IL-6 (*p* < 0.01); compared with the LPT group, rumen fluid in the HPT group diet up-regulated the mRNA expression levels of CCL28 and CCL20 (*p* < 0.05) chemokines in CCL types of BRECs; in addition, compared with the LPT group, rumen fluid in the HPT group up-regulated the mRNA expression levels of CXCL2, CXCL3, CXCL9 and CXCL14 chemokines in CXCL types of BRECs (*p* < 0.01), and the mRNA expression levels of the CXCL5 chemokine tended to be increased (*p* = 0.06).

## 1. Introduction

Carbohydrates are the principal source of dietary energy for ruminants, which can be classified as fiber carbohydrates or non-fiber carbohydrates (NFC). The latter mainly consists of starch, sucrose and pectin, etc. It usually accounts for 60–70% of the dietary proportion of ruminants, mainly providing energy for rumen microbes and hosts while maintaining gastrointestinal health [1,2]. NDF is an important parameter of forage quality, contributing to rumen fill, determining satiety in animals, negatively correlated with feed consumption, and its level greatly impacts rumen health [3]. Increasing concentrations of starch or dietary NFC can raise milk production in dairy cows and is favorable for use of energy and nitrogen in beef cattle output. However, excessive proportions of NFC/NDF may induce the risk of subclinical rumen acidosis, while a lower ratio of NFC/NDF may limit ruminant production performance [4,5]. Changing dietary roughage and concentrate contents is a common method to change dietary structure and nutrient level. However, the ratio of roughage and concentrate contents cannot accurately reflect the contents of different types of carbohydrates in the diet. In contrast, the NFC to NDF ratio can reflect the levels of different types of carbohydrates in quantitative terms. Different NFC to NDF ratios have different effects on production performance, feed utilization efficiency and rumen health of ruminants.

The salubrious breeding of stock and the yield of high-grade animal products need a consistent supply of complete feed [6]. The pelleted TMR is a compound feed, processed by fully mixing pulverized forage and concentrate with various nutritional supplements, granulated and cooled, according to the ruminants’ nutritional requirements for energy, crude protein, crude fiber, vitamins and minerals [7]. The pelleting procedure encompasses steam heating and mechanical strain to break complex fiber constituents and boost starch gelatinization, increasing nutrient digestibility and feed slaking [8]. In comparison with the loose TMR, pelleted TMR has been regarded as an expeditious feed form for ameliorating the digestibility and reducing animal diet sorting, which was a practicable strategy to raise fattening ruminants under the intensive feedlot feeding mode [9]. However, roughage ingredients need to be dried and rough pulverized before they can be used in pelleted TMR.

The sustained high cost of oat grass, alfalfa hay and forage shortage have become the main restricting factors for feedlot yield. As a traditional oil crop, peanut is one of the three edible oil crops in China. Henan is the largest peanut-planting province in China, and has very rich peanut hay resources [10]. Soybean straw is the main by-product of soybean crops, which contains high fiber with indigestible characteristics. It is difficult to be used by monogastric animals, but can be degraded by ruminants to produce volatile fatty acids to provide energy [11,12]. Finding and using these local alternative feeds in pelleted TMR becomes a practical and economical solution to cut feeding costs, while also mitigating the negative socio-environmental effects of agri-industrial waste disposal.

Therefore, it was hypothesized that different proportions of NFC/NDF may be an important factor affecting production performance and rumen health. In particular, the application of pelleted TMR on Holstein growing cattle remains unknown. The objective of this study was to investigate the differences in growth performance, rumen fermentation and inflammatory response in Holstein growing cattle, consuming low or high NFC to NDF ratio pelleted TMR for providing some experimental basis for practical application.

## 2. Materials and Methods

Animal procedures were conducted following the Guidelines for Care and Use Committee of Yangzhou University (Approval No. 20160019).

### 2.1. Processing Requirement of Pelleted TMR

The freshly dried peanut hay and soybean straw were obtained from Zhengyang County Xintiandi Grass Industry Co., Ltd. in Zhumadian city, China. The roughage ingredients including peanut hay and soybean straw were coarsely ground to pass through 8-mm screen (Beijing Grinder Instrument Co., Ltd., Beijing, China), while the concentrate ingredients were ground to pass through 1.2-mm screen. All ingredients were thoroughly mixed. Pelleting at 70 °C, conditioning at 85 °C for 45 s and then forced-air cooling were the pelleting conditions. The compression ratio of the ring die was 1:9.5 (Jiangsu Muyang Group Co., Ltd., Yangzhou, China). The ingredients and chemical compositions of the diet is shown in (Table 1).

### 2.2. Animals and Treatments

Experimentation was conducted at the Ruminant Research Farm of Yangzhou University (longitude 119.46° and latitude 32.78°) from September to November in 2020. Twenty-four healthy castrated male Holstein growing cattle with similar body weight (301 ± 11.5 kg) were enrolled in this study and randomly allocated into two groups (12/pen). Cattle in LPT group were fed basal pelleted TMR with low NFC/NDF ratio (NFC/NDF = 1.07), while the HPT group were fed high NFC/NDF pelleted TMR (NFC/NDF = 1.71), separately. Diets were formulated to meet requirements of NRC (2001) for growing cattle. Throughout the entire duration of the experiment, all Holstein growing cattle were offered ad lib access to clean tap water and pelleted TMR. Holstein growing cattle were fed two times at 17:30 p.m. and 8:30 a.m. every day. Feed allocations were adjusted daily pursuant to the feed refusals (<15%) and the residual’s weight was documented every day before fresh feeding was distributed. The experiment comprised 2 weeks adaptation and 8 weeks fattening duration.

### 2.3. Sample Collection and Analysis

All Holstein growing cattle were weighed fortnightly before morning feeding to calculate the average daily gain. Representative feed samples were collected weekly and immediately stored at −20 °C until further analyses. At the end of the experiment, withers height, body length, and chest girth were measured as quantitative indicators of growth.

With span of 8 h between each time, samples of feces were gathered 3 times a day on the final five days. Feeds and fecal samples were stored at −20 °C until the determination of the apparent digestibility. Acid detergent fiber and neutral detergent fiber were measured by means of Van Soest and a Fiber Analyzer (2000i, Ankom, New York, NY, USA). Crude protein, dry matter (method 930.15), ash (method 942.05) and ether extract (method 920.39) content in samples of offered and refused diet ingredients and feces were analyzed according to AOAC international 2005. Feed consumption behavior was recorded for five consecutive days by a digital video camera (Model C8W 4MP, Hangzhou, China) and counted by playback function.

On the last day of the experiment, two hours after morning feeding, rumen fluid (150 mL) was collected and filtered through 4 layers of cheesecloth for further analysis. The pH value was determined instantly (PB-21, Beijing, China). Further, 10 mL aliquots were stored at −20 °C for the determination of NH_3_-N concentrations following the method of Weatherburn [13]. The concentrations of VFA were tested by utilizing GC Ultra gas chromatograph (Focus GC, Thermo Scientific, Milan, Italy) as depicted by Guo et al. [14]. Microbial protein (MCP) yield was measured following the method that was described by Herejk and Hall [15].

### 2.4. Determination of the Expression Levels of Inflammatory Factors in BRECs by qRT-PCR

Immortalized Bovine rumen epithelial cells (BRECs) were identified and used according to Kang Zhan’s method [16]. Rumen fluid of Holstein growing cattle fed with low NFC/NDF(1.07) and high NFC/NDF(1.71) pelleted TMR was transferred into 50 mL tube and centrifuged at 12,000× *g* at 4 °C for 90 min. Supernatant was carefully absorbed and transferred to 50 mL tube, sterilized with 0.22 μm sterile membrane. In LPT group, 10% rumen fluid of low NFC/NDF(1.07) was added into the medium, while in HPT group, 10% rumen fluid of high NFC/NDF(1.71) was added into another medium, and BRECs were incubated for 6 h for transcriptome analysis, respectively. Rumen fluid of each group was obtained from 8 Holstein growing cattle. qRT-PCR assays were performed using SYBR^®^ Premix Ex TaqTM II Kit (Takara) following the manufacturer’s protocol. The primers used are listed in Table 2. In addition, GAPDH was used to normalize target gene abundance, adopted through calculation of abundance using the 2^−^^ΔΔCT^ method.

### 2.5. Statistical Analysis

The numerical data were analyzed using the independent sample *t*-test. The *p* values are represented in the figures as follows: * *p* < 0.05, and statistical significance was set at *p* < 0.05. The tables present the means and the standard error of the mean (SEM). The calculations were made using SAS 9.4 software (SAS Institute, Cary, NC, USA).

## 3. Results

### 3.1. Effects of Low and High NFC to NDF Ratio Pelleted TMR on Growth Performance and Apparent Digestibility of Holstein Growing Cattle

Final body weight, average daily feed intake, average daily gain, withers height change and body length change in the LPT group were higher (*p* < 0.05) than those of the HPT group, having been increased by 3.79%, 5.71%, 18.70%, 28.57% and 16.67%, respectively. In addition, compared with the HPT group, feed to gain ratio and feed cost (*p* < 0.01) in the LPT group were lowered by 12.24% and 15.35%, respectively. Compared with the HPT group, the LPT group had a tendency to increase CG change (*p* = 0.08). There was no difference in other parameters between the LPT and HPT groups (Table 3).

### 3.2. Effects of Low and High NFC to NDF Ratio Pelleted TMR on Apparent Digestibility of Holstein Growing Cattle

As shown in Table 4, apparent total-tract digestibility of DM and NDF in the LPT group was higher (*p* < 0.05) than those of the HPT group, having been increased by 3.41% and 4.26%, respectively. Compared with the HPT group, the apparent digestibility of ADF in the LPT group was increased by 8.02% (*p* < 0.01). However, there was no significant difference in apparent digestibility of CP between the LPT group and the HPT group (*p* > 0.05).

### 3.3. Effects of Low and High NFC to NDF Ratio Pelleted TMR on Rumination Behavior of Holstein Growing Cattle

As shown in Table 5, the daily feed intake of NDF in the LPT group was higher than that in the HPT group (*p* < 0.05) and the daily rumination time and chewing time in the LPT group were longer than those in the HPT group (*p* < 0.05), but there was no difference in foraging time between the two groups; however, foraging time of DM per kilogram in the HPT group was longer than that in the LPT group (*p* < 0.05). Notably, the foraging time, rumination time and chewing time of NDF per kilogram per day were longer in the the HPT group than in the the LPT group (*p* < 0.05).

### 3.4. Effects of Low and High NFC to NDF Ratio Pelleted TMR on Rumen Fermentation of Holstein Growing Cattle

The effects of low and high NFC to NDF ratio pelleted TMR on fermentation characteristics in the rumen are presented in Table 6. The results showed that compared with the LPT group, the parameters of pH, MCP and C2:C3 (*p* < 0.05) in the HPT group were decreased by 8.57%, 12.46% and 23.71%, respectively. Compared with the LPT group, concentrations of TVFA, C2 and C3 were higher (*p* < 0.05) in the HPT group, and increased by 13.49%, 19.59% and 52.70% respectively. However, it was observed that concentration of NH_3_-N and C4 were not affected by NFC to NDF ratio (*p* > 0.05).

### 3.5. Effects of Low and High NFC to NDF Ratio Pelleted TMR on mRNA Level of Pro-inflammatory Cytokines in BRECs

The mRNA expression levels of IL-1β, IL-6, TNF-α, IL-12, IL-32 and TGF-β were detected by qRT-PCR. GAPDH was used as an internal reference. Rumen fluid in the LPT group and the HPT group came from eight Holstein growing cattle, respectively. As can be seen from the Figure 1, compared with the LPT group, rumen fluid in the HPT group diet up-regulated the mRNA expression levels of BRECs pro-inflammatory factor IL-1β and TNF-α (*p* < 0.05), and meanwhile, up-regulated the mRNA expression levels of BRECs pro-inflammatory factor IL-6 (*p* < 0.01). However, the mRNA expression levels of IL-12 and IL-32 were not changed (*p* > 0.05). There was no significant difference in mRNA expression levels of TGF-β involved in MAPK inflammatory signaling pathway between rumen fluid in the HPT group and the LPT group (*p* > 0.05). These results suggest that rumen fluid in the HPT group diet can promote the pro-inflammatory response of BRECs compared with the LPT group.

From the cluster heat map (Figure 2) obtained by transcriptome analysis, compared with the LPT group, the expression levels of TNF-α, IL-6 and IL-1β were up-regulated in the HPT group (*p* < 0.05), but the expression levels of IL-12, IL-32, TGF-β1 and TGF-β2 were not significantly affected by the HPT group (*p* > 0.05). In addition, the expression levels of IL-24, IL-37 and IL-17D tended to be up-regulated, but there were no statistical differences (*p* > 0.05). The results showed that the results of transcriptome sequencing analysis were basically consistent with the results of qRT-PCR detection, indicating that under the conditions of this test, high NFC to NDF ratio in pelleted TMR will have adverse effects on rumen epithelial cells of Holstein growing cattle, and will promote the expression of pro-inflammatory factors in rumen epithelial cells.

### 3.6. Effects of Low and High NFC to NDF Ratio Pelleted TMR on mRNA Level of Chemokines in BRECs

QRT-PCR was used to detect the mRNA expression levels of chemokines associated with the initial inflammatory response of BRECs. According to Figure 3, compared with the LPT group, rumen fluid in the HPT group diet up-regulated the mRNA expression levels of CCL28 and CCL20 (*p* < 0.05) chemokines in CCL types of BRECs, respectively, but CCL2 was not significantly up-regulated (*p* > 0.05). The mRNA expression of CCL5 chemokine had a tendency to be increased (*p* = 0.07). In addition, compared with the LPT group, rumen fluid in the HPT group diet up-regulated the mRNA expression levels of CXCL2, CXCL3, CXCL9 and CXCL14 chemokines in CXCL types of BRECs (*p* < 0.01), and the mRNA expression levels of the CXCL5 chemokine tended to increase (*p* = 0.06). The results demonstrated that BRECs up-regulated the expression of CCL and CXCL chemokines when fed with the HPT-group diet.

According to the cluster heat map (Figure 4) obtained from transcriptome analysis, compared with the LPT group, rumen fluid in the HPT-group diet could up-regulate CCL20 and CCL28 in CCL type (*p* < 0.05), and tended to up-regulate CCL5. However, the expression levels of CCL1, CCL2, CCL17, CCL22, CCL25, CCL26 and CCL27 chemokines were not affected (*p* > 0.05). Compared with the LPT group, rumen fluid in the HPT group diet up-regulated chemokine expression levels of CXCL2, CXCL3, CXCL9 and CXCL14 in CXCL type (*p* < 0.01), and tended to up-regulate CXCL5, but there was no statistical difference. The above results indicated that the transcriptome sequencing analysis results were consistent with qRT-PCR detection results, indicating that under the conditions of this experiment, high NFC to NDF ratio pelleted TMR will increase the expression of chemokines in the rumen epithelial cells of Holstein growing cattle, and have adverse effects on the rumen health.

## 4. Discussion

Feeding pelleted TMR rather than conventional loose concentrate plus forage to cattle and fattening sheep is an emerging practice in China. Dietary NFC to NDF ratio is often increased in diets for high-energy intake and to maximize beef production or milk yield. However, greater rumen fermentation from feeding more NFC can increase the risk of ruminal acidosis, which can conversely cause disadvantageous impacts on rumen health, production performance and feed consumption [17]. Appropriate NFC to NDF ratio pelleted TMR achieved greater average daily gain, improved feed conversion ratio, increased digestibility of organic matter, acid detergent fiber and neutral detergent fiber in Chinese domesticated black goats [18]. Previous research discovered that growth performance and feed intake were better while the TMR was pelleted, but the digestibility of organic matter, crude protein and acid detergent fiber were not influenced by using pelleted TMR [19]. In this study, our results demonstrated that when the NFC to NDF ratio increased from 1.07 to 1.71, ADFI for LPT animals would consume 4.15 kg NFC and 3.85 kg NDF, while HPT animals would consume 4.77 kg NFC and 2.79 kg NDF. However, the apparent digestibility of DM was significantly higher (75.73%) for animals in the LPT treatment versus the HPT treatment (71.30%). NDF and ADF consumed in the LPT group averaged 2.66 kg and 1.90 kg, while the HPT group averaged 1.86 kg and 1.06 kg. The differences might be on account of a suitable NFC to NDF ratio and an appropriate portion of roughage having been used in the LPT group, which ensure the rumen health of Holstein growing cattle to digest and utilize the nutrients.

Ruminant behavior plays a significant role in the utilization and digestion of roughage by ruminants. It is affected by many factors, such as dietary type, physical structure, feed intake, disease and physiological stage, et al. The NDF content was the main factor affecting rumination time and that dietary NDF intake increased the rumination time [20]. The heifer eating and chewing time per kilogram of dry matter intake were increased with increasing dietary peNDF_8.0_ content, and 18.0% dietary peNDF_8.0_ content was the most suitable for 8–10-month heifers [21]. The results of our research indicated that the daily feed intake of NDF in the LPT group was significantly higher than that in the HPT group and the daily rumination time and chewing time in the LPT group were significantly longer than those in the HPT group, resulting in a higher rumen pH. The results suggested that feeding behavior, especially rumination time, was not affected by low NFC/NDF (1.07) pelleted TMR in this study, but these parameters for high NFC/NDF (1.71) were significantly decreased. This was consistent with the growth performance of Holstein growing cattle.

Rumen pH is a meaningful fermentation index and is affected by a range of factors, including forage to concentrate ratio and feed processing [22,23]. NH_3_-N is the main product of degradation of nitrogen-containing substances, such as proteins, non-protein nitrogen and peptides, and is the primary source of nitrogen that is made use of by rumen microorganisms [24]. VFA is a main energy source for ruminants and could meet 75–80% of the entire energy requirements, of which the molar proportion of yield could be used to describe whether a ration was appropriate for the livestock [25]. Feeding pelleted TMR has a tendency to augment short-chain fatty acid concentration and diminish rumen pH [20]. The ratio of the NFC/NDF (1.66) diet improved digestibility of crude protein and reduced ruminal pH, but decreased ruminal NH_3_-N concentrations in Chinese domesticated black goats [19]. The results of our research demonstrated that ammonia N showed no difference and rumen pH in LPT reflected greater dietary forage and ruminating and chewing to increase saliva and buffer the rumen to higher pH (6.33), compared to HPT with more NFC and lower pH (5.87). The parameters of MCP and C2:C3 in the HPT group were obviously decreased and concentration of TVFA, C2 and C3 were significantly higher in the HPT group. Increased VFA makes sense for HPT because for a more fermentable diet, it follows that acetate and propionate would be higher. These results showed that, in the present study, low NFC/NDF (1.07) pelleted TMR could be used as a fattening diet in 240-day-old Holstein growing cattle. However, high NFC/NDF (1.71) pelleted TMR might have adverse effects on rumen fermentation, which is in line with the ruminating behavior of Holstein growing cattle in our research.

Although the addition of higher NFC in the diet can increase the growth rate or milk production, it may cause some problems, such as variations in microbiota in the rumen environment, and then induce nutrition metabolic disorder, including rumen inflammation, laminitis and intermittent diarrhea [26]. Nevertheless, the rumen epithelium immune system can react to the inflammatory challenge from the lysis of dead bacterial cells in rumen activity. Besides, grain-induced SARA raised the levels of chemokines and cytokines, leading to systemic inflammation. This systemic inflammatory response was related to the disruption of rumen epithelial barrier function induced by diet [27]. Therefore, rumen fluid of low NFC/NDF (1.07) and high NFC/NDF (1.71) pelleted TMR were collected to further explore whether it could actively respond to the immune response of BRECs in vitro. As a multifunctional cytokine, IL-6 is involved in the regulation of immune response, acute response and inflammation [28]. The IL-1β, an important member of the IL-1 family, has attracted much attention due to its important role in inflammation-related diseases. It has strong pro-inflammatory activity and can induce a variety of pro-inflammatory mediators, such as cytokines and chemokines [29,30]. The TNF-α is a multipotent pro-inflammatory cytokine belonging to the TNF ligand superfamily, which plays a diverse role in regulating a variety of developmental and immune processes, including inflammation, differentiation, lipid metabolism and apoptosis [31]. Chemokines are the general term of cytokines that can induce the targeted chemotaxis of immune cells, which can be classified into CCL, CXCL and CX3C type, according to the sequence position of cysteine. CCL chemokines mainly chemotactic monocytes and CXCL chemokines mainly chemotactic neutrophils [32]. In this study, CCL and CXCL chemokines were selected for determination. A previous study showed that rumen epithelial cells induced by LPS can activate the innate immune response by an elevation in chemokines CXCL8 and CXCL6 [33]. The results of our study found that, compared with the LPT group, rumen fluid in the HPT-group diet significantly up-regulated the mRNA expression levels of BRECs pro-inflammatory factor IL-1β, TNF-α and IL-6, suggesting that the HPT group can trigger the pro-inflammatory response of rumen epithelium. Moreover, the HPT-group diet significantly up-regulated the mRNA expression levels of chemokine genes involved in CCL28, CCL20, CXCL2, CXCL3, CXCL9 and CXCL14, indicating that rumen epithelial cells can actively respond to the pro-inflammatory challenge. The rumen epithelial inflammatory factors and chemokine gene expression were up-regulated in dairy cows fed a high-concentrate diet [34], which is consistent with our investigation. These results demonstrated that under the condition of pelleted TMR, high NFC/NDF (1.71) pelleted TMR can have adverse effects on rumen epithelial cells compared with low NFC/NDF (1.07) pelleted TMR.

## 5. Conclusions

We conclude that low NFC/NDF (1.07) pelleted TMR can increase the growth performance, lessen feed costs and promote the digestibility of DM, NDF, as well as ADF of Holstein growing cattle. Moreover, rumen fermentation and inflammatory response were not affected by low NFC/NDF (1.07) pelleted TMR compared with that of high NFC/NDF (1.71). Under the condition of this study, low NFC/NDF (1.07) pelleted TMR is recommended for Holstein growing cattle fattening.

## Figures and Tables

**Figure 1 animals-12-01036-f001:**
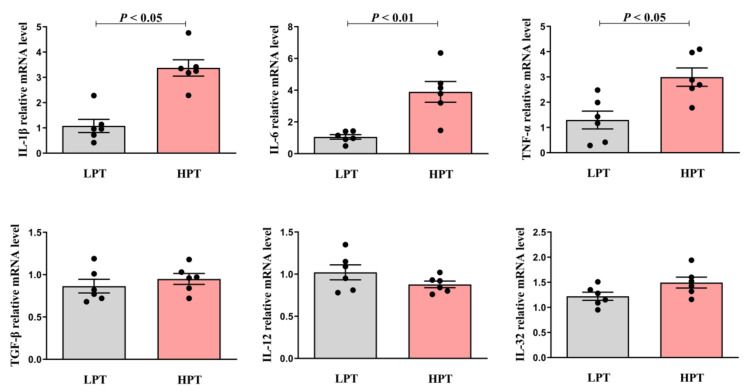
Effects of low and high NFC to NDF ratio pelleted TMR on mRNA level of proinflammatory cytokines in BRECs.

**Figure 2 animals-12-01036-f002:**
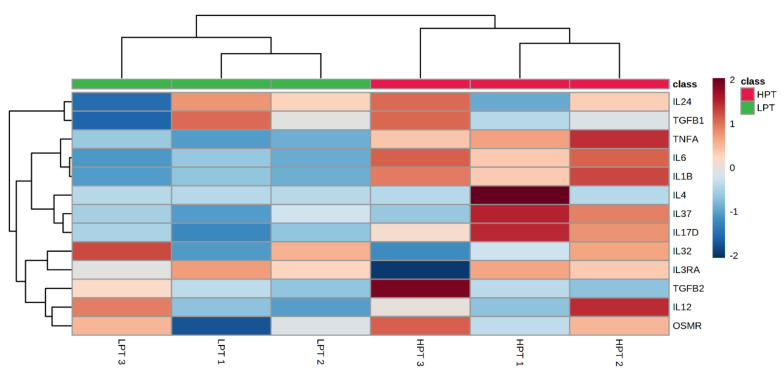
Dendrogram of differential expressed genes involved the pro-inflammatory cytokines. The log_10_ (FPKM+1) value was clustered. Blue represents the low-level genes, and red represents the high-level genes. The color from blue to red represents that the gene expression level is higher and higher.

**Figure 3 animals-12-01036-f003:**
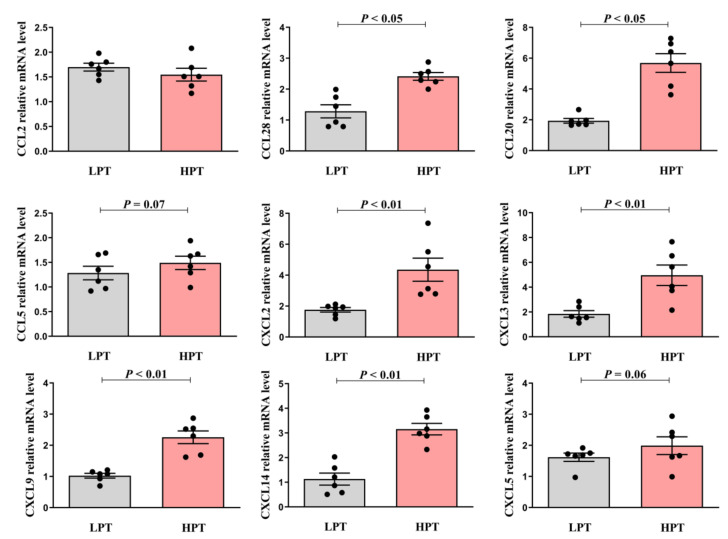
Effects of low and high NFC to NDF ratio pelleted TMR on mRNA level of chemokines in BRECs.

**Figure 4 animals-12-01036-f004:**
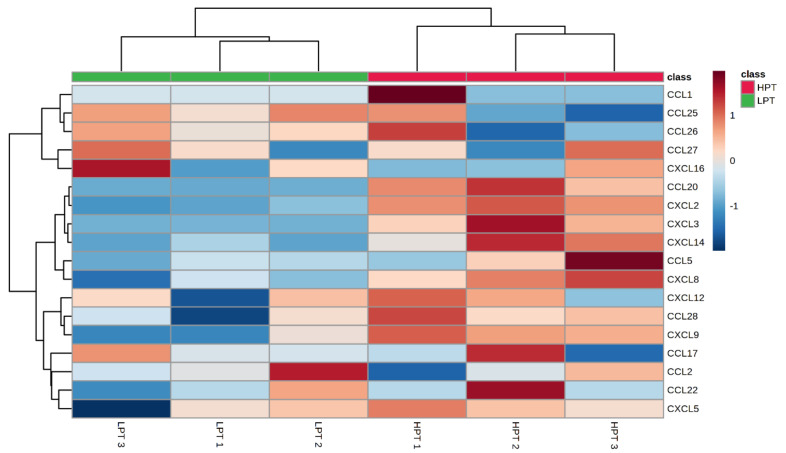
Dendrogram of differential expressed genes involved the chemokines. The log_10_ (FPKM+1) value was clustered. Blue represents the low-level genes, and red represents the high-level genes. The color from blue to red represents that the gene expression level is higher and higher.

**Table 1 animals-12-01036-t001:** Ingredients and chemical compositions of low and high NFC to NDF ratio pelleted TMR (DM basis %).

Ingredients (DM %)	LPT	HPT
Ground corn	23.5	38.5
Wheat bran	5.0	9.0
Soybean meal (46%)	8.0	6.0
Corn germ meal	3.0	4.0
DDGS (corn)	3.0	6.0
Peanut hay	27.0	15.0
Soybean straw	24.0	15.0
Calcium carbonate	1.5	1.5
Calcium phosphate	0.5	0.5
Molasses	2.0	2.0
Salt	0.5	0.5
Premix ^(1)^	2.0	2.0
Nutrient Composition		
Metabolic energy (MJ/Kg) ^(2)^	9.6	11.5
Dry matter (%)	88.2	87.5
Crude protein (%DM)	13.5	13.4
Crude ash (%DM)	11.6	11.8
Ether extract (%DM)	2.8	2.8
Neutral detergent fiber (%DM)	34.7	26.6
Acid detergent fiber (%DM)	25.8	17.8
Non-fibrous carbohydrate (%DM)	37.4	45.4
NFC/NDF ratio	1.07	1.71
Calcium (%DM)	1.20	1.20
Phosphate (%DM)	0.59	0.59

Abbreviations: DDGS, distiller dried grains with soluble; LPT, low NFC/NDF pelleted TMR (NFC/NDF = 1.07); HPT, high NFC/NDF pelleted TMR (NFC/NDF = 1.71). ^(^^1^^)^ The premix per kilogram diet contained 11000 IU VA; 4500 IU VD; 2100 IU VK and 1000 IU VE, 75 mg Zn; 55 mg Mn; 80 mg Fe; 20 mg Cu; 0.5 mg I; 0.2 mg Co; 0.3 mg Se; ^(^^2^^)^ NFC (g/kg DM) =100 − (NDF g/kg + CP g/kg + EE g/kg + Ash g/kg).

**Table 2 animals-12-01036-t002:** Primers for reverse-transcription PCR (RT-PCR) analyses.

Symbol	Primer Sequence, 5′ to 3′	Source	Size (bp)
GAPDH	Sense: GGGTCATCATCTCTGCACCT Anti-sense: GGTCATAAGTCCCTCCACGA	NM_001034034	176
IL-1β	Sense: CAGTGCCTACGCACATGTCT Anti-sense: AGAGGAGGTGGAGAGCCTTC	NM_174093	209
IL-6	Sense: CACCCCAGGCAGACTACTTC Anti-sense: TCCTTGCTGCTTTCACACTC	NM_173923	129
TNF-α	Sense: GCCCTCTGGTTCAGACACTC Anti-sense: AGATGAGGTAAAGCCCGTCA	NM_173966	192
IL-12	Sense: TCTTTTCGGGACTGTTGACC Anti-sense: AAAATCCCCCATCCAAGGTAG	NM_174355	224
IL32	Sense: TCAAGAGAACAGTCCCGAAACC Anti-sense: AGCGTACTTCTTGCTGTGCTTC	NM_005224639	71
TGF-β	Sense: CTGCTGTGTTCGTCAGCTCT Anti-sense: TCCAGGCTCCAGATGTAAGG	NM_001166068	123
CCL2	Sense: GCTCGCTCAGCCAGATGCAA Anti-sense: GGACACTTGCTGCTGGTGACTC	NM_174006	171
CCL5	Sense: CTGCCTTCGCTGTCCTCCTGATG Anti-sense: TTCTCTGGGTTGGCGCACACCTG	NM_ 175827	152
CCL20	Sense: TTCGACTGCTGTCTCCGATA Anti-sense: GCACAACTTGTTTCACCCACT	NM_174263	172
CCL28	Sense: GCTTCTGGAAAGAGTGACAACGT Anti-sense: AGGATGACAGCAGCCAAGTC	NM_001101163	72
CXCL2	Sense: CCCGTGGTCAACGAACTGCGCTGC Anti-sense: CTAGTTTAGCATCTTATCGATGATT	NM_174299	204
CXCL3	Sense: CCCGTGGTCAACGAACTGCGCTGC Anti-sense: AGTTGGTGCTGCCCTTGTTTAG	NM_001046513	217
CXCL5	Sense: CCCGTGGTCAACGAACTGCGCTGC Anti-sense: AGTTGGTGCTGCCCTTGTTTAG	NM_001046513	193
CXCL9	Sense: ACTGGAGTTCAAGGAGTTCCAGCA Anti-sense: TCTCACAAGAAGGGCTTGGAGCAA	NM_001113172	129
CXCL14	Sense: AAGCTGGAAATGAAGCCAAA Anti-sense: GTTCCAGGCGTTGTACCATT	NM_001034410.2	153

**Table 3 animals-12-01036-t003:** Effects of low and high NFC to NDF ratio pelleted TMR on growth performance parameters in Holstein growing cattle.

Growth Parameters	Treatments		SEM	*p*-Value
	LPT	HPT		
BW Initial (kg)	301.0	301.0	11.50	0.45
BW Final (kg)	383.0	369.0	12.35	*p* < 0.05
ADFI (kg/d)	11.10	10.5	0.56	*p* < 0.05
ADG (kg/d)	1.46	1.23	0.09	*p* < 0.05
FI/G	7.60	8.53	0.32	*p* < 0.01
Feed cost (RMB/head/d)	21.5	24.8	1.50	*p* < 0.01
WH: Initial (cm)	130.40	131.20	12.80	0.81
Final (cm)	135.20	136.50	12.30	0.57
Change (cm/d)	0.09	0.07	0.01	*p* < 0.05
BL: Initial (cm)	135.80	137.3	11.3	0.18
Final (cm)	151.60	150.5	13.50	0.35
Change (cm/d)	0.28	0.24	0.02	*p* < 0.05
CG: Initial(cm)	164.05	165.40	14.02	0.36
Final (cm)	172.30	173.50	16.21	0.40
Change (cm/d)	0.15	0.14	0.01	0.08

Abbreviations: BW, body weight; SEM, standard error of the mean; ADFI, average daily feed intake; ADG, average daily gain; FI/G, feed intake to gain ratio; WH, withers height; BL, body length; CG, chest girth.

**Table 4 animals-12-01036-t004:** Effect of low and high NFC to NDF ratio pelleted TMR on nutrient apparent digestibility in Holstein growing cattle.

Nutrients	Treatments		SEM	*p*-Value
	LPT	HPT		
DM (%)	75.73	71.30	1.75	*p* < 0.05
CP (%)	78.56	79.25	4.13	0.16
NDF (%)	69.20	66.37	1.10	*p* < 0.05
ADF (%)	61.03	56.50	1.56	*p* < 0.01

Abbreviations: DM, dry matter; CP, crude protein; NDF, neutral detergent fiber; ADF, acid detergent fiber.

**Table 5 animals-12-01036-t005:** Effects of low and high NFC to NDF ratio pelleted TMR on rumination behavior in Holstein growing cattle.

Behavior Parameters	Treatments		SEM	*p*-Value
	LPT	HPT		
NDFI (kg/d)	3.85	2.79	0.45	*p* < 0.05
FT (min/d)	310.50	306.80	18.35	0.60
DM (min/kg)	27.95	29.21	0.83	*p* < 0.05
NDF (min/kg)	80.65	109.90	10.12	*p* < 0.05
RT (min/d)	371.03	326.50	20.56	*p* < 0.05
DM (min/kg)	33.40	31.10	2.50	0.32
NDF (min/kg)	96.35	117.02	8.61	*p* < 0.05
CT (min/d)	680.10	631.20	28.30	*p* < 0.05
DM (min/kg)	61.27	60.11	2.85	0.79
NDF (min/kg)	176.60	226.20	18.50	*p* < 0.05

Abbreviations: NDFI, neutral detergent fiber intake; FT, foraging time; RT, rumination time; CT, chewing time.

**Table 6 animals-12-01036-t006:** Effects of low and high NFC to NDF ratio pelleted TMR on rumen fermentation parameters in Holstein growing cattle.

Rumen Parameters	Treatments		SEM	*p*-Value
	LPT	HPT		
NH_3_-N (mg/dL)	11.73	12.06	0.26	0.53
Rumen pH	6.33	5.87	0.09	*p* < 0.05
MCP (mg/mL)	3.45	3.02	0.18	*p* < 0.05
Total VFA (mmol/L)	115.38	130.95	5.64	*p* < 0.05
Acetate (C2)/(mmol/L)	59.41	71.05	4.93	*p* < 0.05
Propionate (C3)/(mmol/L)	23.15	35.35	3.65	*p* < 0.05
Butyrate (C4)/(mmol/L)	16.26	16.13	1.10	0.95
C2:C3	2.60	2.02	0.12	*p* < 0.05

Abbreviations: MCP, microbial protein; TVFA, total volatile fatty acids; C2, acetate; C3, propionate; and C4, butyrate.

## Data Availability

The data presented in this study are available on request from the corresponding author.

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
