# Peer review of "Growth Performance, Rumen Fermentation and Inflammatory Response on Holstein Growing Cattle Treated with Low and High Non-Fibrous Carbohydrate to Neutral Detergent Fiber Ratio Pelleted Total Mixed Ration"

_animals, 2022, doi:10.3390/ani12081036_

Round 1
Reviewer 1 Report
The manuscript still requires improvement in the English. The quality of Figure 1 and 3 is still too poor for publication.
Reviewer 2 Report
See attached file please.

Reviewer 3 Report
Please make editorial corrections as suggested in line-by- line attachment

Round 2
Reviewer 2 Report
Very nice. Thank you for making the suggested changes to improve the readability of the paper.
Reviewer 3 Report
Edits have been made satisfying recommendations for improvement. Thank you, and good luck with the rest of the review process
Author Response
Please see the attachment.

This manuscript is a resubmission of an earlier submission. The following is a list of the peer review reports and author responses from that submission.
Round 1
Reviewer 1 Report
The study title suggests that the authors have worked on calves. The animals are 8 months old, so better described as growing cattle. The paper should be proof read to correct error in English. The quality of presentation of Figure 1 and Figure 3 is so poor that it isnt possible to read the text. The discussion needs to be written in a way that it focusses on the present study. At present the discussion is centred around the topic in general.
Some specific corrections:
The abstract should include the aims of the study.
Line 160: define the measurement of body length, point of shoulder to pin bones? Or wither to tail head. Define chest girth - is this a measure of heart girth taken just behind the forelegs?
Line 159 to 166 - were the samples stored in a refrigerator at 4-5oC or in a freezer at -20oC?
Table 4 - what are the units?
Each table needs to have all of the abbreviations defined in the footnotes.
Reviewer 2 Report
Very nice paper. Only one difficulty in interpreting Table 5. Many entries had duplicate units such as DM(min/kg) and NDF(min/kg). It was confusing reading the table and could be presented more clearly.
Reviewer 3 Report
This manuscript is very detailed and technically well written, though somewhat wordy. The science seems soundly developed and detialed, however, there are several instances of improper English throughout the manuscript where articles such as "the" are left out and word usage is out of order. This manuscript would benefit from a native English speaker and writer going through it and editing it for proper word usage and word order. There were too many errors to make it acceptable as it is, and to go through it and show every error will take a long time.
Authors should consider splitting the manuscript into two papers, or at least a paper and a technical note or smaller piece. The biotech portion should be its own paper, as it makes it too lengthy as is. The abstract could then be made shorter and easier to read. Consider starting the abstract out with a sentence telling the importance of the work and the hypothesis, instead of the abrupt beginning it now has.
Authors should consider using a detailed statistical model to more finely tune and differentiate the nutritional findings such as a mixed model, rather than a simple t-test. It is difficult to accept that everything was significantly different and at every level.
The abstract is too lengthy and difficult to follow- suggest simply stating the highest treatment, instead of every percentage difference of every parameter. For example, instead of "(1) final body weight, average daily feed intake, average daily gain, withers height change and body length change in LPT group were significantly higher (p<0.05) than those of HPT group, increased by 3.79%, 5.71%, 18.70%, 28.57% and 16.67%, respectively; " say "body measurements were found to be increased for the LPT group (p<0.05)" and leave the detailed percentages for the paper itself.
Round 2
Reviewer 1 Report
My suggestions are as per my initial feedback. Very little has been done to address the issues with the discussion. Figure 1 and Figure 3 appear to be in the same form as with the initial submission.
